# Defining the Role of the Gut Microbiome in the Pathogenesis and Treatment of Lymphoid Malignancies

**DOI:** 10.3390/ijms24032309

**Published:** 2023-01-24

**Authors:** Shristi Upadhyay Banskota, Sydney A. Skupa, Dalia El-Gamal, Christopher R. D’Angelo

**Affiliations:** 1Division of Hematology and Oncology, University of Nebraska Medical Center, Omaha, NE 68198, USA; 2Fred and Pamela Buffett Cancer Center, University of Nebraska Medical Center, Omaha, NE 68198, USA; 3Eppley Institute for Research in Cancer and Allied Diseases, University of Nebraska Medical Center, Omaha, NE 68198, USA

**Keywords:** lymphoma and gut microbiome, gut microbiome and lymphomagenesis, microbial interventions in lymphoma, lymphoma, gut microbiome

## Abstract

The gut microbiome is increasingly being recognized as an important immunologic environment, with direct links to the host immune system. The scale of the gut microbiome’s genomic repertoire extends the capacity of its host’s genome by providing additional metabolic output, and the close communication between gut microbiota and mucosal immune cells provides a continued opportunity for immune education. The relationship between the gut microbiome and the host immune system has important implications for oncologic disease, including lymphoma, a malignancy derived from within the immune system itself. In this review, we explore past and recent discoveries describing the role that bacterial populations play in lymphomagenesis, diagnosis, and therapy. We highlight key relationships within the gut microbiome-immune-oncology axis that present exciting opportunities for directed interventions intended to shape the microbiome for therapeutic effect. We conclude with a limited summary of active clinical trials targeting the microbiome in hematologic malignancies, along with future directions on gut microbiome investigations within lymphoid malignancies.

## 1. Introduction

The gut microbial genome is confined within the human intestine and is estimated to be 150 times larger than the entire human genome, much of this belonging to bacteria [1,2]. Hence, the gut microbiome is often referred to as a second human genome [1]. The gut microbiome is a collective term for an ecosystem formed by different microorganisms colonizing the human gastrointestinal tract [3]. Microbial dysbiosis is a broad term defined as the loss of “health-promoting” microbes that are commensal to the gut, and/or the deleterious presence of pathogenic ones [4]. This imbalance has been found to be associated with myriad conditions such as chronic inflammatory states, which include inflammatory bowel diseases, infections, and immune dysregulation, resulting in various immune-mediated diseases and neoplastic conditions, including hematologic malignancies [5,6].

In recent years, the study of gut microbiota has received significant attention from researchers attempting to uncover ties between the microbiome and human health. The balance of gut microbiota has been shown to be associated with many cancers and studies are continuing to evaluate their interactions with different cancer treatments. For example, researchers have demonstrated the potentially crucial role of gut microbiota in influencing colon cancer development and progression, through unfavorable microbial community states in the human gut. Specifically, bacteria such as *Bacteroides fragilis* [7,8] and *Fusobacterium nucleatum* [9] are known to induce a pro-inflammatory state in the colon that could potentially enhance oncogenic proliferation in the mucosal microenvironment. There is also a growing body of evidence correlating the role of gut microbiota and hematologic malignancies. The contributions or role of the microbiome may differ with respect to its separate impact on disease pathogenesis and therapy. This review aims to summarize the evolving literature on the relationship of the gut microbiome with lymphomagenesis, along with their role in certain therapies for lymphoma.

Techniques for analysis of the microbiome have advanced over the years. Traditional culture-based approaches have severely limited our appreciation of the genetic depth and diversity within the gut microbiome [10]. This approach has been replaced with sequencing-based approaches, yet the sampling methods used may also influence results. Sampling techniques that have been adapted for microbiome research are summarized in Box 1. Newer techniques for microbiome analysis are further discussed in the Future Directions section.

Box 1Techniques for Microbial SamplingFecal Microbiota: Many microbiome studies use fecal specimen collection [11,12]. The collection of fecal material from study subjects is simple and enables both cross-sectional and longitudinal studies [12]. Recently, the degree to which feces are able to accurately characterize the overall gut microbiota composition, including luminal versus mucosal interfaces, has become a matter of debate.Luminal and Mucosal Microbiota: The luminal microbiota, represented by fecal specimens, is a distinct microbial ecosystem from the mucosa-associated microbiota, represented by colonic/intestinal biopsy samples [13]. The small intestine is enriched with bacteria from the *Proteobacteria* and *Firmicutes* phyla, particularly members of the family *Lactobacillaceae* [14], whereas the colon is dominated by the *Bacteriodetes* and *Firmicutes* phyla.Digesta and Mucosa-associated Microbiota: Bacteria within the digesta differ from the bacteria present in the mucosal inter-folds [14]. The digesta, which refers to the dietary fibers that are currently being digested, is dominated by family members from the *Bacteriodetes* phylum (*Bacteriodaceae*, *Prevotellaceae*, and *Rikenellaceae*) while the inter-folds are enriched with the *Lachnospiraceae* and *Ruminococcaceae* family members of the *Firmicutes* phylum [15]. Optimal sampling methods for these environments remain to be defined.Sample Processing: The use of different methods may influence microbial representation. Recently, a comparison of freezing +/− additives +/− homogenization of samples concluded that overall composition was preserved, but individual taxa may vary [15].

## 2. Microbiome Environments and Lymphomagenesis

Animal models provide a useful tool in determining the intricate relationship between the gut microbiome and health and disease; in particular, the importance of the microbiota in immune development and composition, as well as its role in carcinogenesis [16]. Investigators have demonstrated its significant contribution to disease progression and the eradication of organisms such as *Staphylococcus aureus*, with improved outcomes in cutaneous T-cell lymphoma (CTCL) [17]. Tegla et al. also demonstrated the increased colonization of *Staphylococcus* spp. via CTCL microbiome analysis, when compared with healthy controls. This suggests a potential role of the dominance of *S. aureus*, leading to direct enhanced antigen presentation, enhanced clonal expansion of T cells, and the upregulation of pro-inflammatory cytokines, contributing to disease progression in CTCL [17]. Broad-spectrum antibiotics used in the treatment of advanced-stage CTCL lesions were associated with tumor regression and a reduction in the fraction of malignant T cells, suggesting that the microbiota can contribute to lymphoma pathogenesis [18].

Primary lymphomas that recapitulate the features of gastric mucosa-associated lymphoid tissue lymphoma (gastric MALT lymphoma) are known to arise in the stomach [19,20]. Epidemiologically, gastric MALT lymphoma is closely associated with the infection of a specific bacteria, *Helicobacter pylori* [21]. While *H. pylori* is present in approximately 50% of the world’s population [21], the incidence of gastric MALT lymphoma is low, suggesting that there are specific mechanisms by which *H. pylori* invades the gastric mucosa and, subsequently, evades the host immune system. The first study to investigate and further elucidate this link was conducted by Wotherspoon et al. In this study, the investigators showed that *H. pylori* infection significantly increased the risk of gastric MALT lymphoma, due to the vast majority of these patients being infected with *H. pylori* [19]. Furthermore, this study observed that the lymphoid follicles in these *H. pylori*-infected individuals developed into lymphoid tissues that were morphologically identical to gastric MALT lymphoma [19]. In 1993, Wotherspoon et al. further characterized the role of *H. pylori* in gastric MALT lymphoma using six patients, all of whom showed histological and genetic evidence of gastric MALT lymphoma. Remarkably, treatment with antibiotics led to the eradication of *H. pylori* infection and the subsequent regression of the lymphoma in the majority of the study population [22]. Since then, *H. pylori* infection and the different mechanisms contributing to the genesis of gastric MALT lymphoma have been revealed. Tumor-infiltrating T cells that are stimulated by *H. pylori* antigens have been shown to enhance the growth of B cells in gastric MALT lymphoma [23]. Additionally, another study showed that antigen-dependent development and the subsequent progression of gastric MALT lymphoma may be triggered by the expression of the CD40 ligand, in concert with Th2 cytokines, IL-4, and IL-10, by activated T cells [24]. These studies highlight the intricate communication between microbes, T cells, and B cells in the pathogenesis of MALT lymphoid malignancies and suggest that the balance of immune-activating and immune-suppressing cell populations and their signals contributes to the unique features in MALT lymphoma, including susceptibility to antibiotic-driven strategies.

Specific tumor-infiltrating T cells and B cell receptor (BCR) signaling also play a role in the pathogenesis of gastric MALT lymphoma. In a study conducted by Craig et al., investigators used a murine model of *H. felis*-induced gastric MALT lymphoma to show that the development of MALT lymphoma requires both BCR signaling, via the poly-reactivation of tumor-derived immunoglobulins (Igs) with self-antigens, and tumor-infiltrating CD4+ T cells [25]. Moreover, most of these CD4+ T cells were FOXP3+ regulatory T cells that were being recruited by the tumor cells via chemokines such as CCL17 and CCL22 [25]. The in vivo inhibition of FOXP3+ regulatory T cells indeed resulted in the regression of gastric MALT lymphoma [25]. In line with this finding, Garcia et al. showed a higher FOXP3+/CD3+ cell ratio in *H. pylori*-positive gastric MALT lymphoma than in *H. pylori*-negative gastric MALT lymphoma [26]. Additionally, the expression of CD86, a co-stimulatory molecule that activates B cells to proliferate and produce IgG in B-cell lymphoma, was shown to be significantly associated with the sensitivity of *H. pylori*-dependent gastric MALT lymphoma to *H. pylori* eradication [27]. As illustrated in Figure 1, we demonstrate the intertwined relationship involving the gut microbiome, lymphomagenesis, lymphoma-directed therapies, and microbial interventions, details of which will be given later in this article.

## 3. Immune-Mediated Mechanisms of Lymphoma Development and Therapy

### 3.1. Dendritic Cell Activation

Dendritic cells (DCs) have long been characterized as professional antigen-presenting cells that work by recognizing pathogen-associated molecular patterns (PAMPs) via receptors such as the toll-like receptors (TLRs), allowing them to bind to pathogens such as bacteria [28]. Moreover, DCs are at the interface of the innate and adaptive immune system [28,29]. In general, intestinal DCs will encounter the gut flora in two different ways. In the first path, commensal bacteria or other intact antigens gain access to GALT via transcytosis across specialized enterocytes, known as microfold or M cells [28,29]. Here, the DCs capture the rogue bacteria or antigens in the Peyer’s patches, which then migrate to mesenteric lymph nodes, where they are recognized by the pattern recognition receptors (PRRs) [28,29]. The second way in which intestinal DCs encounter bacteria or antigens assumes that the DCs are directly sampling the symbiotic bacteria and antigens that are already present within the intestinal lumen, constantly deciphering between beneficial and detrimental microbes [28]. Recently, the relationship between intestinal DCs and tumors has been extensively studied. Sivan et al. demonstrated that the supplementation of *Bifidobacterium* to mice engrafted with melanoma led to DC activation and enhanced anti-melanoma T-cell responses [30]. While certain DC subsets have been considered as predictive markers for gastric cancers [28], more research is needed to thoroughly cover their role in hematological malignancies such as lymphoma.

### 3.2. T Cell Activation

An important indirect mechanism through which the gut microbiome affects lymphomagenesis is via immune system alterations, also known as aberrant immune responses. Intestinal microbiotas communicate with and influence the gut’s immune system, in order to maintain harmony between immune responses, namely, immune tolerance and immune activation [31]. Recognition of each microbe’s molecular pattern (pathogen- or microbiome-associated molecular pattern) inside the host gut occurs through pattern recognition receptor systems (PRRs) such as toll-like receptors (TLRs), which are expressed in the intestinal epithelial cells, along with intestinal macrophages and dendritic cells [32]. Upon the recognition and invasion of the microbe into the epithelium, an appropriate immune response is then activated. This constitutes the activation of myriad intracellular signaling pathways, such as kinases, adaptor molecules, and transcription factors, ultimately triggering gene expressions for the synthesis of multiple cytokines (both pro-inflammatory and anti-inflammatory), chemokines, cell adhesion molecules, and immunoreceptors [33]. While common commensal bacteria help in reducing phagocyte migrations in the gut epithelium and aid in the formation of a protective mucosal layer [34], pathogenic microbes initiate a pro-inflammatory cascade by initiating a systemic change in the immune cell population. They activate DCs to secrete pro-inflammatory cytokines, such as IL-8, IL-12, and IL-23, which, in turn, facilitate T-cell activation through the differentiation of naïve T cells into Th1 and Th17 cells [16].

Cozen et al. performed an interesting analysis of the timing of exposure to microbiomes in the lifetime of young adults and its correlation with the occurrence or development of young-adult Hodgkin’s lymphoma (HL) [35]. To gather exposure information, the group obtained information from each eligible twin living pair, with at least one member having a diagnosis of young-adult HL. The group explored the exposure history of young adult HL-case twins and compared that with their unaffected control twins (lymphoma-free). This study concluded that behaviors likely to increase early oral exposure to microbiomes and its subsequent influence on the production of helper T cell cytokines were associated with a decreased susceptibility to HL development, compared to behaviors associated with exposure to microbiomes later in life [35]. Examples of some exposures that were associated with significant results in the study included appendectomy, smoking, eczema, and more frequent behaviors associated with increased oral-oral or fecal-oral exposures.

A recent study published by Andrlova et al. explored the mechanism underlying diversity in intestinal microbiota leading to improved outcomes after allogeneic hematopoietic stem-cell transplantation (allo-HSCT). They demonstrated the increased prevalence of unconventional regulatory T cells, such as innate-like mucosal-associated invariant T cells (MAIT cells) and Vd2 cells, in people with preserved intestinal diversity. This, in turn, correlated with favorable overall survival and less acute graft-versus-host disease (GVHD) events [36]. Such unconventional cells have long been recognized as potent early T-cell responders when exposed to metabolites from intestinal microbiomes via antigen presentation, and help to interlink innate and adaptive immunity [37]. This is also consistent with prior translational studies that have demonstrated the association of MAIT cells with better outcomes after allo-HSCT [38,39,40]. Early immune reconstitution with Vd2 cells in the first couple of months after allo-HSCT has also been associated with reductions in acute GVHD, in addition to improved survival outcomes [41]. Commensal microbial diversity helps to regulate the maintenance of unconventional T cells, thereby preserving their immunological interplay with microbe-derived ligands, together supporting favorable outcomes after allo-HSCT [36].

Calcinotto et al. explored the effect on Th17 cells, the gut microbiome, and pathogenesis in a mouse model of multiple myeloma [42]. Th17 cells have been associated with disease progression, with advanced bony lesions and the clonal expansion of plasma cells in the bone marrow (BM) of people with multiple myeloma [43]. Pathogenic species such as *Prevotella heparinolytica* were found to be associated with earlier progression in the Vk Myc myeloma mouse model, particularly by enhancing Th17 cell activation within adjacent Peyer’s patches along the gastrointestinal tract [42,44]. Interestingly, these Th17 cells were found to migrate to the BM, where, in an eosinophil-driven mechanism, they promoted the progression of multiple myeloma into an active disease state. Ultimately, these data highlight a potential direct immunological link between the gut and the BM tumor microenvironment (TME), which, in turn, influences disease progression in multiple myeloma [42].

With our increasing understanding of the microbiome’s contribution to the host immune system, emerging studies are now exploring the regulatory roles of gut microbiota in the treatment of cancers via their effects on the immune system [45,46]. Indeed, the gut microbiome is recognized as an important determinant of the response to immune checkpoint blockade in cancer treatment [46].

### 3.3. Effects of Gut-Associated Metabolites

Over the past decade, microbiome research has highlighted the concept that the eukaryotic host and its intestinal microbiota have co-evolved to orchestrate multiple aspects of host physiology and aid the host in maintaining a stable microbial community [47,48]. Multiple mucosal defenses in the intestinal lumen maintain the compartmentalization of the microbiota within the lumen, keeping an interface between the microbe-free host, the microbiota, and the environment [47,48,49]. Recently, a growing number of studies have also suggested that the microbiome plays a critical role in host immune system development and differentiation, as well as host metabolic homeostasis [48].

One striking example of the role of the gut microbiota and its importance in immune system development is the observation that germ-free mice, which are born and raised in the absence of any microbial colonization, have profoundly underdeveloped immune systems [50]. A study conducted by Chung et al. showed that this feature of germ-free mice can be largely reversed by the adult colonization of host-specific microbiota. Here, colonizing germ-free mice with mouse microbiota or mouse-segmented filamentous bacteria partially restored the organisms required for full immune maturation and conferred protection against certain bacterial infections [51].

While these two studies provide important insights into the role of microbial communities in immune system development, they also highlight the importance of having a host-specific microbial configuration that allows host immunity to function properly. This function is, in large part, due to the ability of different microbial configurations to produce, modulate, and degrade a large array of molecules known commonly as metabolites [47,48,49].

Metabolites provide functional complementation to the host, aid in microbiota–host communications via innate immune receptors, and can act as mediators of bacterial signaling to drive changes in the composition and function of the microbiota [47]. Most metabolites originate in three ways: (a) diet-dependent microbial products that are directly linked to our diet, such as short-chain fatty acids (SCFAs); (b) diet-independent microbial products that are synthesized de novo by gut microbes, such as lipopolysaccharides, a structural component of Gram-negative bacteria [49]; (c) metabolites that are produced by the host and are biochemically modified by gut microbes, such as secondary bile acids, a product of the intestinal microbiota metabolism of host-produced primary bile acids [52]. In fact, around 10% of metabolites in the blood and more than 50% in feces and urine are derived from or modified by the gut microbiota [53,54]. As such, recent studies have shifted their focus to functionally characterizing and determining the role of gut-derived metabolites in modulating the homeostasis and function of both innate and adaptive immunity. Metabolites of particular importance in lymphomagenesis include butyrate, an SCFA, and the metabolites of nitrogen-recycling bacteria [53,54].

Dietary fibers and non-digestible carbohydrates are integral components of the human diet; however, humans lack the enzymes to degrade these components, and, as a result, commensal bacteria in the cecum and colon ferment these fibers [55]. The resulting products of this bacterial fermentation are SCFAs (primarily acetate, propionate, and butyrate). Butyrate generated by the gut microbiota (e.g., *Firmicutes*) has demonstrated contradictory effects, as it was found to not only facilitate the generation of regulatory T cells but also have the ability to down-regulate pro-inflammatory cytokines via influencing intestinal macrophages by inhibiting histone deacetylases [55]. Specifically, in lymphoma, Wei et al. demonstrated that microbial fermentation from a high-fiber diet increased butyrate concentrations in the blood and at tumor sites. Furthermore, the increase in butyrate due to these high-fiber diets elicited protective effects by exhibiting anti-proliferative and pro-apoptotic activity in both mouse and human lymphoma cells [55]. Similarly, another study found that the major commensal bacteria affected in primary intestinal lymphoma (PIL) belonged to the *Eubacterium* genus, a bacterium often localized in the human gut that is characterized by its butyrate-producing abilities. Here, it was demonstrated that *E. rectale* could prevent the genesis of intestinal lymphoma by alleviating chronic inflammation and attenuating B cell responses to intestinal bacteria, thereby affecting downstream TLR4/MYD88/NF-kB signaling [56].

Another interesting avenue is the potential role played by the metabolites produced by nitrogen-recycling bacteria. One such study aimed to characterize how alterations of the gut microbiome in multiple myeloma patients influence disease progression [57]. Using shotgun metagenomic sequencing of human fecal samples from multiple myeloma patients and healthy controls, the investigators found significant differences in the bacterial composition and metabolite profiles between multiple myeloma patients and healthy subjects [57]. Namely, nitrogen-recycling bacteria, such as *Klebsiella* and *Streptococcus* species, along with the metabolite L-glutamine were highly enriched in multiple myeloma patients, which was presumably driven by host-secreted nitrogen in such forms as urea or ammonia [57]. This study concluded that the alterations seen in multiple myeloma patients resulting from excessive host-secreted nitrogen altered the gut microbiome in ways that allowed for the production and fermentation of metabolites that subsequently promoted myeloma progression.

## 4. Clinical Implications of the Gut Microbiome in Lymphoma Diagnosis and Therapy

### 4.1. Lymphoma/Myeloma Diagnosis

Emerging studies are attempting to identify the specific microbial signature that would contribute most to the tumorigenesis and proliferation of chronic lymphocytic leukemia (CLL) B cells. An example is a recent, small study conducted in Denmark, in which CLL patients were compared to healthy controls, attempting to understand the differences in their gut microbial composition. This study demonstrated lower diversity in the overall gut microbiome and reported an increased abundance of specific organisms, such as *Bacteroides* and *Proteobacteria* species, suggesting a possible role of the immune activation and dysfunction induced via such organisms in individuals with CLL. The study also found a lesser abundance of bacteria producing short-chain fatty acids, such as *Lachnospiraceae* and *Ruminococcaceae*, in patients with CLL [58]. Similar findings identifying lower alpha bacterial diversity in the gut microbiome in patients with multiple myeloma compared to healthy controls have also been reported [59]. Finally, a separate study on extranodal natural killer (NK)/T cell lymphoma discovered that specific gut microbial compositions are associated with NK/T cell lymphoma diagnosis and are enriched in NK/T cell lymphoma patients compared to healthy controls; the presence of specific taxa was identified, including a *Streptococcus parasanguinis–Romboutsia timonensis* index that can be associated with prognosis [60].

### 4.2. Chemotherapy/Immunotherapy 

The microbiome environment prior to treatment with chemo-immunotherapy in different types of lymphoma, such as diffuse large B cell lymphoma, marginal zone lymphoma, follicular lymphoma, etc., has been identified as a significant predictor of treatment outcome. Higher pre-treatment microbial diversity and overall distinct microbial composition, with the differential abundance of certain microbial taxa, were found more commonly in responders when compared to non-responders. As an example: an increased prevalence of organisms such as *Dorea formicigenerans* and *Faecalibacterium prausnitzii* was associated with better treatment responses [61]. These findings are also consistent with prior studies in melanoma patients, which illustrated the finding that the response to immunotherapy is modulated by the gut microbiome [46,62]. The commensal microbiome may, in fact, control the response of cancer to chemotherapy by regulating the TME and so-called inflammatory tone of the disease site [63]. This effect was demonstrated in a study conducted by Pflug et al., wherein patients with relapsed lymphoma ultimately had lower overall response rates and reduced progression-free survival with concurrent exposure to anti-Gram-positive antibiotics while receiving cisplatin chemotherapy [64].

Another promising study, utilizing machine learning in patients with non-Hodgkin’s lymphoma (NHL) who were undergoing HSCT, developed a bloodstream infection risk index based on the gut microbiome composition prior to allo-HSCT. The predictivity of future infection risks through the given scoring system demonstrated a sensitivity of 90% and a specificity of 90%, thus implying that the pre-transplant gut microbiome could be used as an important predictor of post-transplant infection risk [65]. Cyclophosphamide is an example of one of the alkylating chemotherapies used in the first-line treatment of CLL. The anti-neoplastic mechanism of the drug has been found to be linked with gut microbiome composition in animal studies [66,67]. A retrospective study performed by Pflug et al. explored the chemotherapeutic efficacy of cyclophosphamide in individuals with CLL who were exposed to Gram-positive organism-specific antibiotics such as vancomycin. They demonstrated a significant reduction in progression-free survival in people exposed to anti-Gram-positive antibiotics for the same dose intensity of chemotherapy, compared to those without antibiotic exposure [64].

Prior studies in solid tumors have recently revealed that the previous or concurrent use of antimicrobial agents might decrease the treatment efficacy of immune checkpoint inhibitor (ICI) therapy, suggesting that the possible disruption of the gut microbiota with antibiotics might lead to the impairment of cytotoxic T cell responses against tumor cells [68,69,70,71]. With growing evidence of the promising efficacy of immunotherapy in classical relapsed/refractory HL, the effect of effective antibiotic use on outcomes in people with HL receiving ICI therapy was evaluated in a single institution study [68]. Prior use of antibiotics within 90 days of starting ICI therapy in people with HL was found to be associated with a negative impact on outcomes such as median survival and progression-free survival. Additionally, researchers have also proposed a positive association between the preservation of gut microbiome diversity and the sensitivity of multiple myeloma tumor cells to different therapeutic options, including cyclophosphamide [67] and allogeneic stem cell transplantation [72].

### 4.3. Autologous Stem Cell Transplantation

Recipients of autologous stem cell transplantation (auto-HSCT) are likely to be subjected to gut microbial dysbiosis via similar mechanisms to those experienced by people undergoing allo-HSCT: exposure to anti-microbial therapy, nutritional modifications, and intestinal mucosal injuries from high-dose chemotherapy [73,74]. D’Angelo et al. reported that the loss of bacterial diversity in the peri-engraftment period following auto-HSCT was associated with the response to transplant. The peri-engraftment period in this study was defined as 2–7 days before melphalan chemotherapy pre-transplant and the first day of neutrophil engraftment (defined as an acute neutrophil count higher than 500 cells/µL post-engraftment) [75]. Khan et al. demonstrated that the degree of injury to the gut microbial ecosystem and the subsequent dysbiosis is comparable to that seen in allo-HSCT recipients [76]. The patterns of domination by specific bacterial taxa also occurred in a similar fashion in both auto- and allo-HSCT recipients. A subsequent follow-up performed by the team to explore the clinical impacts of peri-engraftment diversity disruption on the fecal microbiota samples of people undergoing auto-HSCT demonstrated unfavorable overall survival and progression-free survival rates in patients with lymphoma, multiple myeloma, and amyloidosis [76]. Although the mechanism remains to be elucidated, the use of melphalan as the conditioning chemotherapeutic agent for auto-HSCT, an alkylating agent similar to cyclophosphamide, allows us to speculate that a similar mechanism of microbiota translocation to the mesenteric lymph nodes and spleen-activating T cells, as demonstrated for cyclophosphamide, may be occurring [63].

### 4.4. Allogeneic Stem Cell Transplantation 

The loss of intestinal microbial diversity with the use of antibiotics at engraftment was independently correlated with greater mortality over the 3 years following allo-SCT. Patients with lower microbial diversity died more often due to infections or GVHD than those patients with higher microbial diversity [77]. This finding was later supported by a large multi-center study with 1362 patients, which demonstrated higher transplant-related mortality and higher mortality secondary to GVHDs in patients with lower gut diversity upon an analysis of pre-transplantation patient fecal specimens. Lower gut diversity was found to be associated with the proliferation of a single bacterial taxon leading to an increased likelihood of bloodstream infection, also contributing to increased deaths [78].

The expansion of studies Involving functional shifts in the gut microbiome and its clinical impacts after allo-HSCT have also demonstrated an association between the gut microbiome and pulmonary complications post-engraftment. Interestingly, the domination of members from the *Gammaproteobacteria* taxa post-engraftment was found to be predictive of pulmonary complications and overall death [79].

### 4.5. CAR T Cell Therapy

CD19-directed chimeric antigen receptor (CAR) T-cell therapy represents a breakthrough treatment option in heavily pre-treated relapsed/refractory CD19-positive lymphomas, with the potential for durable remission [80]. Despite the transforming revolution seen with anti-CD19 CAR T-cell therapy, with response rates up to 40% compared to 10% with standard chemotherapies in the relapsed/refractory disease setting, adverse events such as ICANS (immune effector cell-associated neurotoxicity syndrome) and CRS (cytokine release syndrome) with CAR T-cell therapy raises concerns, calling for strategies to mitigate them [81]. Additionally, about three-fifths of the patients could still relapse or remain refractory despite CAR T-cell therapy. Smith et al. evaluated the role of the intestinal microbiome on therapy outcomes such as relapse/non-response and therapy-related toxicities post-CAR T-cell therapy. Prior broad-spectrum antibiotic exposure to medicines such as piperacillin/tazobactam, meropenem, and imipenem/cilastatin in the four weeks before CAR T-cell therapy resulted in unfavorable survival outcomes (both overall and progression-free survival) and increased cytokine-related toxicities such as ICANS. The increased incidence of ICANS after CAR T-cell infusion with prior antibiotic exposure was found to occur across the use of any costimulatory domains for CAR T-cell infusion [82]. The increased correlation of CAR T-cell therapy with ICANS could possibly be linked to potential gut microbiome attributes/factors affecting blood-brain barrier permeability, also known as the gut–brain axis, which finding is consistent with similar conclusions from animal studies [83]. Interestingly, the study also evaluated the association between responses to treatment and certain species of bacteria. For example, microbial species within the class *Clostridium* were correlated more strongly with a complete response at day 100, whereas greater occupancy of the species *Veillonellales* was associated with lower complete response rates at day 100 [82].

### 4.6. Gut-Microbiome-Directed Interventions

One of the best clinical examples of successful microbial intervention is the utilization of a fecal microbiota transplant (FMT) after recurrent/refractory *Clostridium difficile* infection [84]. Another field of emerging interest is that of using dietary modifications to enhance gut microbiota. Prebiotics are defined as nutritional supplements that support beneficial bacterial species, and one such study evaluating a prebiotic approach to target the microbiome in patients undergoing autologous stem-cell transplantation (SCT) is currently enrolling at the University of Nebraska Medical Center. This study aims at preserving gut microbial diversity in multiple myeloma and lymphoma patients who are undergoing autologous SCT (NCT05135351). Other studies are evaluating the use of prebiotic therapy in autologous SCT, as well as the effect of different antibiotic therapies (NCT03078010) or GVHD regimens (NCT03959241) on the microbiome when used during allogeneic SCT.

Current investigational microbial approaches include the utilization of live therapies, such as FMT or probiotics, and dietary interventions such as nutritional supplements, prebiotics, and dietary modifications. In Table 1, we highlight selected studies involving the gut microbiome in connection with hematologic malignancies. The list is not comprehensive but is intended to display the growing interest in microbial-directed therapies to treat hematologic malignancies.

## 5. Future Directions

Over the past decade, the advent of newer techniques for studying the microbiome has enabled enormous progress in understanding the human gut microbiome. These new technologies have enabled us to address some of the prior limitations in microbiome analysis noted above. Amplicon sequencing is one of the most widely used techniques of partial genome sequencing; it utilizes PCR primers, such as 16S rRNA fragments from bacteria and internal transcribed spacers from fungi, in order to only target a specific region of a gene [12]. At present, 16S rRNA gene sequencing is limited by its restriction to bacteria taxa and genus-level resolution. Shotgun metagenomics is another commonly used emerging technique in microbiome research, which enables complete genomic study via DNA sequencing at greater depth (beyond 16S) and informs about the complete list of microbial taxa present in a microbiome. Other examples of novel microbial analytics techniques are: (i) metatranscriptomics, a technique for transcribed RNA sequencing; (ii) metaproteomics, a technique for mass spectrometric protein analysis in a sample; and (iii) metabolomics, a technique to study small-molecule metabolites [12]. The concept of “multi-omic” analysis to integrate these large datasets for a better understanding of the human–microbiome environment is emerging. The term “holobiont” is another emerging idea with, perhaps, more clinical implications for microbiome studies. Park et al. explored the intricate genetic, microbial, and immunological inter-linking seen in gastric cancer by utilizing a multi-omic approach to improve a detailed and deeper understanding of the microbiome and human gene expression in various stages of gastric pathophysiology, such as the healthy, gastritis, and carcinoma stages [85].

Recognition of the gut microbiome as a new emerging and enabling cancer hallmark mirrors the increased knowledge and appreciation of its substantial role in various types of cancers [86,87], including lymphoid malignancies. With ongoing studies continuing to improve our understanding of the gut microbiome and its considerable role in lymphomagenesis and therapy outcomes, the current challenge is to incorporate this knowledge into well-controlled disease-specific mechanistic studies and, ultimately, clinical practice, with the potential manipulation of the microbiota-gut-lymphoma axis as a target for future translation studies.

Although considerable progress has been made in the last few years toward understanding the mutually beneficial interactions of the gut microbiome and host, our mechanistic understanding of how it operates remains incomplete. The microbial community remains deeply complex and poses analytic challenges; terms such as “favorable” and “unfavorable” bacteria warrant further clarification. Additionally, landmark studies have demonstrated a disparity in terms of informing which exact species/members of the gut microbiome are the most important drivers in lymphomagenesis, and it is unclear whether targeted therapies focusing on one specific series are even going to provide meaningful results. The majority of the studies performed in the field of lymphoid diseases and the gut microbiome, even when well-characterized, are limited by a small sample size. The cost associated with newer-generation sequencing techniques is only one price to pay for understanding the microbiome genome. Other critical dimensions to consider when designing and interpreting gut microbiome studies in patients with lymphoid malignancies include previous therapies, co-existing malignancies/diseases, gender, age, diet, and weight. Another emerging factor to examine is the similarities and/or differences observed across races/ethnicities. Microbiota differences across populations that vary in terms of ethnicity and lifestyle have been observed by Brooks et al. In a comprehensive study of the connections between ethnicity and differences in gut microbiota, the investigators observed differences in microbiota composition between ethnicities [88]. Nonetheless, there is still limited understanding regarding the etiology of these differences, where genetics, culture and lifestyle, and diet may all contribute to the observed differences in the gut microbiome between races and ethnicities [88]. Substantial research is still crucial to determine how best to modify the microbiome, with dedicated studies addressing distinct disease settings, such as different subtypes of lymphoma, as well as the demographics of the patient, to provide an opportunity for a more personalized approach. However, it is probably worth concluding that starting antibiotics without a substantial infectious work-up in people with lymphomas and other B-cell malignancies such as multiple myeloma, especially around the time of malignancy-related therapy administration, should probably be reconsidered.

Lastly, the tumor microbiome (i.e., the microbiome present within tumor tissues) in lymphoid malignancies has yet to be explored. Its existence, distribution, and composition in tumor sites and/or TME niches are unknown. Investigations to elucidate the tumor microbiome on cancer development, immunity at tumor sites, and if it correlates to clinical features in lymphoid malignancies are warranted [89].

Despite these challenges, our understanding of the gut microbiome’s influence on host immunity and, in turn, oncogenesis and therapeutics is growing significantly. Gut microbiome research in lymphoid malignancies presents an exciting new frontier for ongoing clinical investigation, given the rapid rise of immunotherapies, and may one day become an important aspect of personalized medicine to treat patients suffering from lymphoma.

## Figures and Tables

**Figure 1 ijms-24-02309-f001:**
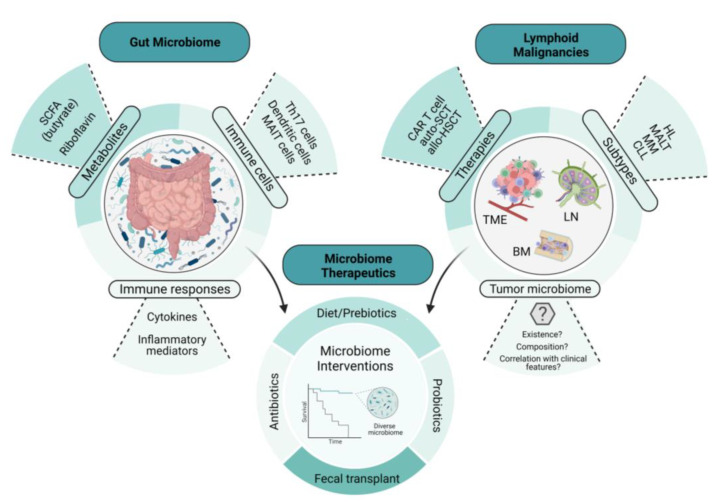
The intertwined relationship between the gut microbiome, lymphomagenesis, lymphoma-directed therapies, and microbiome-directed interventions. This illustration depicts the relationship between the gut microbiome, lymphoid malignancies, and microbiome therapeutics. The examples of microbiome therapeutics that are provided demonstrate how these approaches change key features and readouts of the gut microbiome. These changes in the microbiome not only influence lymphoid malignancies directly but also affect the therapies used to treat them, including immune cellular therapies such as CAR T-cell therapy, potentially through the emerging role of the tumor microbiome. SCFA, short-chain fatty acid; CAR T cell, chimeric antigen receptor T cell; auto-SCT, autologous stem-cell transplant; allo-HSCT, allogeneic hematopoietic stem-cell transplant; HL, Hodgkin’s lymphoma; MALT, mucosa-associated lymphoid tissue; MM, multiple myeloma; CLL, chronic lymphocytic leukemia; TME, tumor microenvironment; BM, bone marrow; LN, lymph node.

**Table 1 ijms-24-02309-t001:** Select diet/microbiome-related clinical trials on hematologic malignancies.

Brief Study Title	Disease/Population	Intervention	Status
Choosing the best antibiotic to protect the gut microbiota during stem-cell transplant	Allo-HSCT for any hematologic malignancies	Piperacillin-tazobactam versus cefepime for the treatment of neutropenic fever	Recruiting (NCT03078010)
Prebiotics during ASCT for lymphoma/myeloma (PRIMAL)	Allo-SCT for lymphoma /myeloma	Resistant potato starch	Recruiting (NCT05135351)
Plant-based diet for MGUS and smoldering myeloma	Monoclonal gammopathy of undetermined significance or smoldering myeloma	Plant-based diet	Recruiting (NCT04920084)
Effects of prebiotics on the gut microbiome in patients undergoing HSCT (HCTDiet)	Allo-HSCT for lymphoma	Prebiotic foods	Recruiting (NCT04629430)
Dietary manipulation of the microbiome-metabolomic axis for GVHD mitigation	Allo-HSCT for lymphoma	Resistant potato starch	Recruiting (NCT02763033)
Intermittent fasting and CLL/SLL	Chronic lymphocytic leukemia or small lymphocytic leukemia	Intermittent fasting	Active, Not recruiting (NCT04629430)
Fecal microbial transplant after allogeneic stem cell Transplantation	Allo-HSCT for any hematologic malignancies	FMT	Not yet recruiting (NCT04935684)
A novel vaccine as monotherapy or combination therapy in indolent NHL	Follicular and marginal zone lymphoma	Tumor-antigen or microbiome peptide vaccine	Recruiting (NCT04669171)
Safety and efficacy of curcumin in children with ALL (CurCumPedALL)	Acute lymphoblastic leukemia	Curcumin	Recruiting (NCT05045443)

## Data Availability

Not applicable.

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
