# Peer review of "Defining the Role of the Gut Microbiome in the Pathogenesis and Treatment of Lymphoid Malignancies"

_ijms, 2023, doi:10.3390/ijms24032309_

Round 1

Reviewer 1 Report

There are no points of concern in the central part of this manuscript.
Moreover, the cited and their discussion are presented in good style. However, authors have some minor points that need to be addressed.

I am interested in the relationship between gut microbiota and inflammatory diseases and the treatment of refractory diseases including FMT.

Therefore, I found this manuscript very interesting to read.

So, one question: are there any racial differences in the relationship between gut microbiota and disease in the literature discussed in this manuscript or in the authors' knowledge?

If so, do you know if it is due to culture, diet, or phenotypic variation due to the human genome, etc?

If there are racial differences or other factors, why not introduce them in the manuscript as an aspect of personalized medicine?
If not, there is no need to mention it.

Author Response

Reply: Thank you very much for your comments and suggestions.  We have addressed them as follows:

Minor points:

  1. So, one question: are there any racial differences in the relationship between gut microbiota and disease in the literature discussed in this manuscript or in the authors’ knowledge? If so, do you know if it is due to culture, diet, or phenotypic variation due to the human genome, etc? If there are racial differences or other factors, why not introduce them in the manuscript as an aspect of personalized medicine? If not, there is no need to mention it.

Reply:  

Thank you for bringing this observation to our attention.  Based on review of the literature, to our knowledge there are no racial disparities at play in the relationship between gut microbiota and lymphoma. As such, this was not discussed within our manuscript. However, to address the emerging research surrounding the relationship of race/ethnic disparities in relationship to gut microbiome research we have added the following sentences to the section 5. Future Directions of our manuscript as seen below in red text:   

“Other critical dimensions to consider when designing and interpreting gut microbiome studies in patients with lymphoid malignancies include previous therapies, co-existing malignancies/diseases, gender, age, diet, and weight. Another emerging factor to examine is the similarities and differences observed across races/ethnicities. Microbiota differences across populations differing in ethnicity and lifestyle have been observed by Brooks et al. In a comprehensive study of the connections between ethnicity and differences in gut microbiota, investigators observed differences in microbiota composition between ethnicities (30513082) (Reference:92). Nonetheless, there is still limited understanding regarding the etiology of these differences in the gut microbiome between races and ethnicities (30513082) (Reference:92). Substantial research is still crucial to determine how to best modify the microbiome with dedicated studies addressing distinct disease settings such as different subtypes of lymphoma as well as demographics of the patient to provide an opportunity for a more personalized approach. However, it is probably worth concluding that starting antibiotics without substantial infectious work-up in people with lymphomas and other B-cell malignancies such as multiple myeloma especially around the time of malignancy related therapy administration should probably be reconsidered.”

Reviewer 2 Report

Understanding the gut microbiome is an important topic in human health. The authors reviewed the roles of the gut microbiome in the context of lymphoid malignancies. The highlighted gut microbiome-immune-oncology axis is an unique perspective. The authors summarized the relationship between gut microbiome and lymphomagenesis, as well as active clinical trials targeting the microbiome in hematologic malignancies. This review is informative for future studies in the field. I have no significant comments on the manuscript, with the exception of a few comments below.

The Figure 1 caption can be more descriptive, highlighting the key information to be conveyed.

Some genus and species names are not italicized, such as those in Box 1.

Please also check proper citation references and adjust all to the journal rules.

Author Response

Comments and Suggestions for Authors: Understanding the gut microbiome is an important topic in human health. The authors reviewed the roles of the gut microbiome in the context of lymphoid malignancies. The highlighted gut microbiome-immune-oncology axis is an unique perspective. The authors summarized the relationship between gut microbiome and lymphomagenesis, as well as active clinical trials targeting the microbiome in hematologic malignancies. This review is informative for future studies in the field. I have no significant comments on the manuscript, with the exception of a few comments below.

Reply: Thank you for reviewing our manuscript and providing your suggestions to enhance the work. We have addressed your comments and suggestions and indicate the changes as follows:

Minor points:

  1. The Figure 1 caption can be more descriptive, highlighting the key information to be conveyed.

Reply: Thank you for this suggestion and we agree that a more descriptive caption for Figure 1 was needed.  We have added the following caption to Figure 1:

“This figure depicts the relationship between the gut microbiome, lymphoid malignancies, and microbiome therapeutics. The examples of microbiome therapeutics provided demonstrate how these approaches change key features and readouts of the gut microbiome. These changes in the microbiome not only influence lymphoid malignancies directly, but also therapies used to treat them including immune cellular therapies like CAR T cell therapy, potentially through the emerging role of the tumor microbiome. SCFA, short-chain fatty acid; MAIT cells, mucosal associated invariant T cells; CAR T cell, chimeric antigen receptor T cell; auto-SCT, autologous stem cell transplant; allo-HSCT, allogeneic hematopoietic stem cell transplant; HL, Hodgkin’s lymphoma; MALT, mucosa-associated lymphoid tissue; MM, multiple myeloma; CLL, chronic lymphocytic leukemia; TME, tumor microenvironment; BM, bone marrow; LN, lymph node.”

  1. Some genus and species names are not italicized, such as those in Box 1.

Reply: We thank the reviewer for pointing this out. We have italicized the genus and species names within Box 1 and have reviewed the remainder of the manuscript, italicizing any rogue genus and species names.

  1. Please also check proper citation references and adjust all to the journal rules.

Reply: Thank you for bringing this to our attention. We have reviewed all references and confirm that the citations are in compliance with the journal rules and will work with the editorial staff as needed moving forward.